# Less than One-shot: Named Entity Recognition via Extremely Weak Supervision

**Letian Peng** and **Zihan Wang** and **Jingbo Shang***

University of California, San Diego

{lepeng, ziw224, jshang}@ucsd.edu

## Abstract

We study the named entity recognition (NER) problem under the *extremely weak supervision* (XWS) setting, where only one example entity per type is given in a *context-free* way. While one can see that XWS is *lighter than one-shot* in terms of the amount of supervision, we propose a novel method X-NER that can outperform the state-of-the-art one-shot NER methods. We first mine entity spans that are similar to the example entities from an unlabelled training corpus. Instead of utilizing entity span representations from language models, we find it more effective to compare the context distributions before and after the span is replaced by the entity example. We then leverage the top-ranked spans as pseudo-labels to train an NER tagger. Extensive experiments and analyses on 4 NER datasets show the superior end-to-end NER performance of X-NER, outperforming the state-of-the-art few-shot methods with 1-shot supervision and ChatGPT annotations significantly. Finally, our X-NER possesses several notable properties, such as inheriting the cross-lingual abilities of the underlying language models. [1]

## 1 Introduction

Named Entity Recognition (NER) with Extremely Weak Supervision (XWS) is the task of performing NER using only one entity example per entity type as the supervision. As shown in Table 1, 1-shot NER is arguably the closest setting to XWS, however, XWS shall be viewed as even weaker supervision, making it very compelling when the annotation is difficult or expensive.

To develop effective NER models under the XWS setting, the keystone is to find sufficiently large, high-quality pseudo labels from raw texts to train the NER model. A straightforward solution is to adapt few-shot NER methods (with uncertainty quantification) to select confident predictions

Table 1: Different NER setups. We only show one entity type "Artist" here for simplicity. Our XWS setting shall be considered weaker than 1-shot since the example entity is given in a context-free way.

| Setup | Example Supervision from Human |
|---|---|
| FEW-SHOT | I love the works by [Davinci]Artist. I'm a big fan of [Van Gogh]Artist's art. Paintings by [Monet]Artist are breathtaking. [Michelangelo]Artist's incredible sculptures. |
| ONE-SHOT | I love the works by [Davinci]Artist. |
| XWS (1-ENT) | Artist: Davinci |

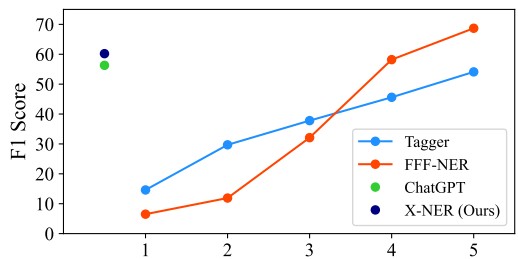

Figure 1: Recent SOTA few-shot NER methods fall dramatically as the available shots decrease on CoNLL03, much weaker than our X-NER under the XWS setting.

on raw texts as pseudo labels. However, existing few-shot NER works typically achieve decent accuracy with a default setting using 5 shots (Huang et al., 2020); when the number of shots approaches 1, their performance drops catastrophically. Concretely, we vary the number of shots for a recent state-of-the-art (SOTA) few-shot NER method FFF-NER (Wang et al., 2022) and a standard sequence labeling NER tagger (Devlin et al., 2019) and show their performance in Figure 1. Under the standard $F_1$ metric in the CoNLL03 dataset, they only achieved under-20 $F_1$ scores in the 1-shot scenario, much lower than the desirable 60 $F_1$ they can achieve with 5-shots. We argue that such low performance is likely inevitable for methods that involve fine-tuning language models (LMs) on the limited training data, which is a common practice in strong performing few-shot NER methods (Yang and Katiyar, 2020; Fu et al., 2021; Cui et al., 2021;

---

*Corresponding author.

[1] Our code is released at KomeijiForce/X-NER

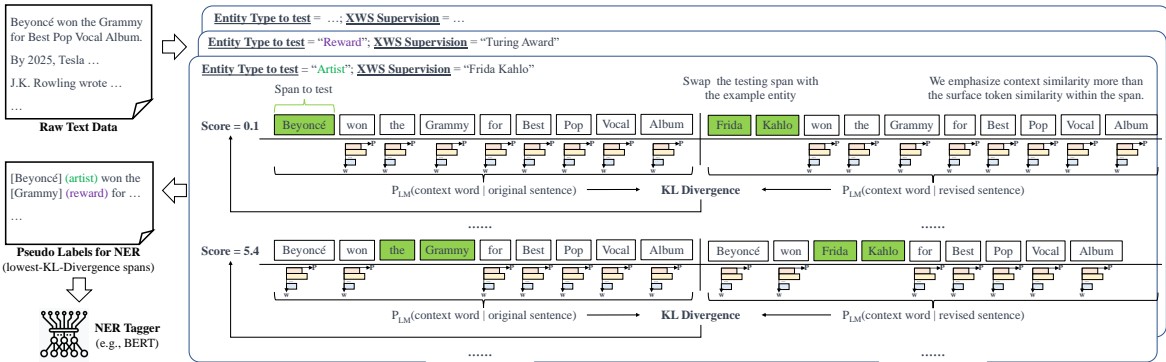

Figure 2: An overview of X-NER.

Ma et al., 2022). Therefore, we propose to obtain high-quality pseudo labels using the pre-trained LM's intrinsic understanding of entities, without any fine-tuning. We need to find a metric (based on the LM) to quantify the similarity of a span in an unlabelled corpus to a target example entity.

It is tempting to directly calculate a span's similarity with an entity based on their LM representations, however, this approach faces two challenges: (1) entities are likely phrases of different lengths and it is non-trivial to define a fixed-length vector representation for multiple tokens, and (2) context understanding is crucial for the generalization capability of an NER model, however, the LM representations of an entity or a span emphasizes more on their surface tokens (Gu et al., 2021).

In this paper, we propose X-NER as shown in Figure 2. Specifically, to remedy both challenges, we propose a span test, which measures the change of the span's context when the span is swapped with the example entity. A lower change indicates that the span is of a higher similarity to the entity example. To quantify the change, we consider the context token-by-token and gather the LM's prediction distribution of the tokens before and after the swap. Precisely, for a masked language modeling trained language model, the prediction distribution is obtained by replacing the token with a mask and obtaining the LM's masked prediction probabilities; for a casual language modeling (CLM)-based LM, we consider all text before the desired token, and we gather the LM's prediction of that token. We then measure the Kullback–Leibler (KL) divergence between the prediction distributions before and after replacing the span. We average the divergence over the context (only tokens to the right of the span for CLM to obtain the final quantification of the change. We apply a corpus-level ranking among all spans and select top-$K$ entities to construct the pseudo-labels. We finally train a

standard sequence labeling NER tagger as a proof-of-concept, while we note that noise-robust NER methods might bring even better performance.

We conduct extensive experiments to compare our X-NER with various few-shot NER methods and ChatGPT for direct inference or pseudo data annotation. Remarkably, X-NER outperforms all these methods with a relatively small LM (i.e., RoBERTa) as the backbone. Further analyses show that X-NER has several interesting properties: (1) it favors bidirectional masked LMs over similarly sized unidirectional CLM-based LMs, (2) it scales well with the capabilities of language models, and (3) it inherits the cross-lingual abilities of the underlying language models.

Our contributions are summarized as follows.

- We explore the challenging Extremely Weakly Supervised NER problem, and achieve substantially higher performances than previous SOTA few-shot methods.
- We propose to use a distributional difference of an entity span's context when it is replaced with an entity example, which is much more effective than the span's representation itself.
- We empirically studied our method extensively. Besides strong performances, we additionally demonstrate its suitable language model types (prefer Masked LM over Causal LM), the effect of language model scaling (the larger the better), and its inherited cross-lingual NER ability.

## 2 Related Work

### 2.1 Few-shot NER

Current studies on few-shot Named Entity Recognition (NER) can be divided into two categories: recognition from scratch and transferring from other domains. The latter category, known as domain transferring NER (Huang et al., 2022; Li and Qian, 2023; Han et al., 2023), trains models on an-

notated NER datasets and then uses meta-learning to learn new entity labels with limited data.

Our work focuses on NER from scratch and there are two main approaches in this field. The first approach involves prompting by templates (Cui et al., 2021; Lee et al., 2022), while the second approach involves aligning NER with pre-training in the learning objective (Paolini et al., 2021; Wang et al., 2022). These two approaches utilize either the definition of entities or unsupervised learning objectives to outperform traditional sequence tagging models. Unfortunately, with 1-shot or even less supervision, their performances suffer from overfitting and are not different from the tagger. To address this issue, our method applies training-free mining to collect entities with high quality and then trains the NER model with them.

## 2.2 Extrememly Weak Supervision

Extremely Weak Supervision (XWS) refers to the setting where a model is expected to solve a certain task with only the minimal supervision that a human requires to solve the task. It was initially proposed for text classification where extremely weak supervision would require the model to classify texts into classes when the only information is the class name of each class (Wang et al., 2021b). The typical workflow of XWS in text classification involves aligning texts to classes via Seed Matching (via surface forms or representations) on unlabelled text and then training a supervised model on such pseudo-annotated dataset (Meng et al., 2020; Park and Lee, 2022; Zhang et al., 2021; Zhao et al., 2022). Language model (zero-shot) prompting is another solution to XWS tasks, however, has been shown to be sub-par against these pseudo-dataset training approaches which can effectively learn on an unsupervised training set (Wang et al., 2023). Our work is the first that addresses NER in an XWS setting, where prior methods usually require about 5-shot sentence annotations per class. We show that our method can achieve reasonable performance with only one single example entity per class.

## 2.3 PLM-based Metric

In recent years, several metrics have been proposed that leverage PLMs as metrics. BERTScore (Zhang et al., 2020) computes token-level F1 scores between generated and reference texts using contextualized word embeddings from BERT, achieving a better correlation with human judgments compared to traditional metrics like BLEU. BLEURT (Sellam

et al., 2020) is another metric developed by Google Research that uses a fine-tuned BERT model to predict human evaluation scores on a combined dataset of existing evaluation resources.

Explicit predictions from pre-trained language models (PLMs) have been widely used for various tasks beyond just generating implicit word representations. Perplexity, which is a measure of the PLM's language modeling ability, has been applied not only for evaluating language models, but also for natural language inference (Holtzman et al., 2021), question answering (Holtzman et al., 2021; Kumar, 2022), and text compression (Niu et al., 2019). Recently, Peng et al. propose to use divergence between masked prediction as semantic distance. We adapt this idea to span test and implement entity mining with a high precision.

## 3 Preliminary and Definitions

**Named Entity Recognition** is a sub-task of information extraction that seeks to locate and classify named entities in text into pre-defined categories. Given a sentence $X = [x_1, \cdots, x_n]$ where $x_i$ is the $i$-th token in the sentence, and $n$ is the total number of tokens. The objective of the NER task is to identify entities in the sentence. For a specific entity type, each entity is represented by a pair of boundaries $(i, j)$ where $x_{i:j} = [x_i, \cdots, x_j]$ signifies that the tokens from $i$-th to $j$-th, inclusive, form a named entity of that type.

**Pretrained Language Models** such as BERT (Devlin et al., 2019), GPT (Radford et al., 2019), and others (Lewis et al., 2020; Raffel et al., 2020) learn to predict the distribution of a token $x_i$ based on its context $X$ through their training objective, represented as $f_{LM}(x_i|X)$. The context, contingent on the model's architecture, could encompass tokens around $x_i$ in BERT-like Masked Language Modeling (MLM) where $x_i$ is replaced by a special mask token during training, or just the preceding tokens in casual language modeling (CLM) models such as GPT. The distribution $f(x_i|X)$ reflects the probable occurrence of tokens in the vocabulary at position $i$ given the context and is an effective dense representation of the context information. We will use this distribution to tell the difference between two contexts $X$ and $X'$.

## 4 Our X-NER

In this section, we introduce our method X-NER to train a NER model with a raw corpus and given

only one entity example per type as supervision.

## 4.1 Overview

We show an illustration of X-NER in Figure 2. We start with XWS Supervision and extract model annotated, or called pseudo, entities on the raw text data. We refer to the model[2] as a miner. Precisely, for each pseudo-annotated sentence $\hat{X} = [\hat{x}_1, \cdots, \hat{x}_n]$, there are several $(i, j)$ pairs identified by the miners that signify pseudo entities $\hat{x}_{i:j} = [\hat{x}_i, \cdots, \hat{x}_j]$. After that, a new NER model is trained on the pseudo-annotated data. The major reason of needing a pseudo-annotation step, instead of directly serving the miner as the final model, is of inference efficiency. This pseudo-annotation then training step abstracts the latency of the miner as pre-processing, therefore, potentially slow miners can still be used efficiently.

## 4.2 Context-Aware Entity Span Ranking

For pseudo NER labeling, our objective is to extract spans $x_{i:j} = [x_i, \cdots, x_j]$ from an unlabeled sentence $X = [x_1, \cdots, x_n]$ that are most similar to a seed entity $Z = [z_1, \cdots, z_m]$. Thus, the span test is designed to evaluate the similarity $S(\cdot)$ of each possible span $x_{i:j}$ to $Z$ conditioning on the context $X$.

$$s_{i,j}^{(X,Z)} = S(x_{i:j}, Z|X).$$

We extract entities by selecting the spans with similarity scores $s_{i,j}^{(X,Z)}$ above a threshold value.

**Cosine Similarity: A straightforward solution.** As a traditional metric for vector representation similarity, we first use cosine similarity to introduce the workflow of our span testing, and to provide a reasonable baseline.

$$S_{cos}(x_{i:j}, Z|X) = \\ \text{Cosine}(\text{Enc}(x_{i:j}|X), \text{Enc}(Z|X_{[i:j \to Z]})).$$

Specifically, we first use the PLM encoder $\text{Enc}(\cdot)$ to incorporate context $X$. To calculate $\text{Enc}(x_{i:j}|X)$, we get the output representations from the $i$-th to $j$-th elements of $\text{Enc}(X)$ from a language model, and then apply mean-pooling (Wang et al., 2021a) over the tokens to get a fixed length representation. For $\text{Enc}(Z|X_{[i:j \to Z]})$, we replace the $i$-th to $j$-th

---

[2]In our case, our span-test; in other cases, it could be a NER model trained elsewhere or a generative model that answer NER annotation prompts

tokens in $X$ with $Z$, and retrieve and average token representations for $Z$ similarly as what we did for $\text{Enc}(X)$.

**Our proposed Divergence Distance: A more context-aware solution.** While cosine similarity is a straightforward method, it is not necessarily consistent with the primary objective for which the PLM encoder $\text{Enc}(\cdot)$ is optimized for. We propose a divergence distance evaluation inspired by Peng et al. (2023) that accounts the language modeling objective more. The core insight is that instead of comparing the spans $Z$ and $x_{i:j}$, which may have different lengths, directly, we compare their impact on a shared context. Since language models capture contextualized information, their context would reflect nuanced differences between the two spans. Formally, we decompose the similarity as

$$s_{i,j}^{(X,Z)} = \sum_{k \in 1:n \setminus i:j} -D((x_k|X), (x_k|X_{[i:j \to Z]})),$$

where D is a function that calculates the difference of a token $x_k$ in context $X$ versus $X_{[i:j \to Z]}$, after replacing the testing span with the seed entity; and, the negative sign translates difference to similarity.

Recall that in Section 3 we defined $f_{LM}(x_i|X)$ as a language model's prediction of token $x_i$ for the context $X$. The prediction is a distribution over the vocabulary, which is a nuanced quantification of the context. Therefore, we employ a standard KL Divergence on the language model predictions as the distance metric $D$. That is,

$$s_{i,j}^{(X,Z)} = - \sum_{k \in 1:n \setminus i:j} \text{KL}(f_{LM}(x_k|X) || f_{LM}(x_k|X_{[i:j \to Z]}))$$
$$= - \sum_{k \in 1:n \setminus i:j} \sum_{v \in V} p(x_k = v|X) \log \frac{p(x_k = v|X)}{p(x_k = v|X_{[i:j \to Z]})}$$

**Two-way testing w/ annotation-free context.** Our span testing can be further extended to a two-way testing, where the seed span is provided together with a seed context $Y$ where $Z = Y_{u:w}$. We use the label name of the seed span to create an annotation-free context background to enable the two-way testing: "The [ENTITY LABEL]: [SEED SPAN]". We don't view this as additional supervision, but rather akin to an instruction template used in prompting methods.

The similarity $s_{i,j}^{(X,Z)}$ can be thus transformed into the following expression $s_{i,j,u,w}^{(X,Y)}$ with the seed context $Y$.

$$s_{i,j,u,w}^{(X,Y)} = s_{i:j}^{(X,Y_{u,w})} + s_{u:w}^{(Y,X_{i,j})}$$

For the $s_{i:j}^{(X,Y_{u,w})}$ side, we only calculate two neighboring elements $k \in \{i-1, j+1\}$ to regularize the imbalance in sentence length, which are verified to be effective according to Peng et al. (2023). We avoid using mean-pooling, as the decay of word interdependency in long sentences could inadvertently cause the metric to favor short sentences. For the $s_{u:w}^{(Y,X_{i,j})}$ side, as the tests on the seed context use the same neighboring elements, $s_{u:w}^{(Y,X_{i,j})}$ calculates all neighbors in the annotation-free context. Thus, the two-way testing adds $|Y| - |Z|$ calculations to each span test.

## 5 Experiments

### 5.1 Datasets

To verify the effectiveness of our X-NER, we conduct experiments on four NER datasets ranging from general to specific domains.

- **CoNLL03** (Sang and Meulder, 2003) is a prominent NER corpus on news corpus that encompasses four general entity types: person (PER), location (LOC), organization (ORG), and miscellaneous (MISC).
- **Tweebank** (Jiang et al., 2022) shares the same four entity types as CoNLL03 on tweet messages.
- **WNUT17** (Derczynski et al., 2017) centers on emerging and rare entity types in user-generated content, such as social media posts.
- MIT **Restaurant** (Liu et al., 2013) features a variety of entities in the restaurant domain, such as dish names, restaurant names, and food types.

A summary of the datasets along with corresponding dataset statistics can be found in Appendix B.

When mining entity spans from unlabeled training sets, for all compared methods that need candidate spans, we focus on spans with a length of no more than 3 tokens and exclude any containing stop words. [3]

### 5.2 Compared Methods

We denote our method as **X-NER**. The default version of our X-NER is divergence distance-based 2-way span testing with the annotation-free context technique. We also include a 1-way variant which only test the seed entity as an ablation study. Under the XWS setting, our major baselines, which all share the same pseudo dataset training pipeline as X-NER, are as follows.

---

[3]Empirically, this simple rule can boost the efficiency while enjoying a $90\%$ recall of all ground-truth entities, except $83.4\%$ for Restaurant.

- **PromptMine** ranks entities by the language model prompting probability using "*X, $X_{i:j}$ is a/an <mask> entity.*" for MLM or "*X, $X_{i:j}$ is a/an*" for CLM. This method only requires the label names of the entity types, which has **less supervision than XWS**, that requires an additional example entity. We denote this setting as **L**.
- **InstructMine** queries `gpt-3.5-turbo` (OpenAI, 2023) with an instruction to annotate entities. All seed spans and templates are included in Appendix A.
- **X-NER w/ Cosine** is a variant of X-NER that uses cosine similarity of the span and the entity mentioned in Section 4.2.

We also compare with few-shot NER methods. Since they can not operate on the XWS setting, we provide them with **1-shot** supervision, which is **more supervision than XWS**.

- **ETAL** (Chaudhary et al., 2019) is a method with pseudo-labeling to search for highly-confident entities that maximize the probability of BIO sequences.
- **SEE-Few** (Yang et al., 2022) expands seeded entities and applies an entailment framework to efficiently learn from a few examples.
- **TemplateNER** (Cui et al., 2021) utilizes human-selected templates decoded by BART for each sentence-span pair during training.
- **EntLM** (Ma et al., 2022) employs an MLM pre-trained model to predict tokens in the input sentence, minimizing the gap between pre-training and fine-tuning but potentially introducing context-related issues.
- **SpanNER** (Fu et al., 2021) separates span detection and type prediction, using class descriptions to construct class representations for matching detected spans, though its model designs differ from the backbone pre-trained model BERT.
- **FFF-NER** (Wang et al., 2022) introduces new token types to formulate NER fine-tuning as masked token prediction or generation, bringing it closer to pre-training objectives and resulting in improved performance on benchmark datasets.
- **SDNET** (Chen et al., 2022) pre-trains a T5 on silver entities from Wikipedia and then fine-tunes it on few-shot examples.

In addition to the few-shot NER methods in previous research, we additionally consider two baselines that can help us ablate X-NER:

- **PredictMine** follows a similar procedure as X-NER where it also first makes pseudo annotations

Table 2: NER results under XWS. $L$ implies using label names only.

| | Method | Model | CoNLL03 | | | Tweebank | | | WNUT17 | | | Restaurant | | |
|---|---|---|---|---|---|---|---|---|---|---|---|---|---|---|
| | | | P. | R. | F. | P. | R. | F. | P. | R. | F. | P. | R. | F. |
| $L$ | PromptMine | GPT2-XL | 18.3 | 18.8 | 18.5 | 0.5 | 0.2 | 0.3 | 1.7 | 1.4 | 1.5 | 24.8 | 13.9 | 17.8 |
| | | RoB-L | 23.7 | 20.4 | 22.0 | 0.4 | 0.1 | 0.2 | 1.2 | 0.4 | 0.6 | 21.7 | 11.1 | 14.7 |
| XWS | InstructMine | ChatGPT | 61.9 | 51.6 | 56.3 | 23.5 | 39.7 | 29.6 | 22.1 | 18.0 | 19.8 | 29.3 | **35.5** | 32.1 |
| | X-NER | RoB-L (Cos) | 37.7 | 20.3 | 26.4 | 6.1 | 3.1 | 4.1 | 19.5 | 6.2 | 9.4 | 19.1 | 7.9 | 11.2 |
| | | RoB-L (Div) | **71.4** | **52.1** | **60.2** | **50.4** | **40.5** | **44.9** | **51.5** | **34.6** | **41.4** | **58.8** | 27.3 | **37.3** |

Table 3: Comparison between our method's XWS performance and previous few-shot method's **1-shot** performance on CoNLL03. ∗: Methods with pseudo-labeling. †: SDNET uses a language model pre-trained on silver entities from Wikipedia.

| | Method | P. | R. | F. |
|---|---|---|---|---|
| XWS | X-NER | 71.4 | 52.1 | 60.2 |
| 1-shot | Tagger | 17.2 | 12.8 | 14.6 |
| | PredictMine* | 13.5 | 17.2 | 15.1 |
| | CertMine* | 20.2 | 14.7 | 17.0 |
| | ETAL* | 26.4 | 17.1 | 20.8 |
| | SEE-Few | 21.5 | 26.7 | 23.8 |
| | TemplateNER | 12.6 | 6.7 | 8.8 |
| | EntLM | 13.6 | 29.0 | 18.5 |
| | SpanNER | 56.1 | 3.8 | 7.1 |
| | FFF-NER | 39.1 | 3.6 | 6.5 |
| | SDNET† | 56.6 | 44.1 | 51.3 |
| 5-shot | Tagger | 58.9 | 55.1 | 56.9 |
| | FFF-NER | 70.2 | 65.3 | 67.7 |

Table 4: Precision@1000 of different entity ranking methods on CoNLL03.

| Method | Setup | PER | LOC | ORG | MISC |
|---|---|---|---|---|---|
| PromptMine | GPT2-XL | 7.3 | 32.2 | 28.4 | 3.4 |
| | Rob-L | 6.7 | 40.1 | 35.1 | 0.1 |
| X-NER | Cos-2 way | 27.1 | 41.0 | 9.0 | 15.1 |
| | Div-1 way | 88.6 | 64.1 | 24.2 | 12.9 |
| | Div-2 way | **98.4** | **97.9** | **62.3** | **82.2** |

`gpt-3.5-turbo`[4], X-NER has a slight edge on CoNLL03 and Restaurant, while much better on Tweebank and WNUT17. We suspect that this indicates a preference for X-NER in noisy user text. Finally, the results also reveal that using the cosine similarity of averaged token representations between the entity and the span as the metric leads to much worse performance. This justifies the importance of our divergence distance metric.

Table 3 presents the comparison of X-NER to methods that require 1-shot data. Due to a limitation of resources, we run the experiments only on the standard CoNLL03 benchmark. X-NER outperforms all previous few-shot baselines by a very significant gap, indicating its substantially stronger ability in handling NER under extremely weak supervision. We also include 5-shot results of a vanilla tagger and the SOTA few-shot method FFF-NER. We find the X-NER can rival strong 5-shot NER methods while requiring much less supervision (5 fully annotated sentences per entity type v.s. one example entity for each entity type). This result also implies that there should be much more room for few-shot NER methods to improve. Finally, PredictMine and CertMine have a similar (poor) performance as prior methods, indicating that the performance advantage of X-NER is not mainly because of mining on an unlabelled corpus, but rather our insights on using the divergence metric.

on an unlabeled corpus, and then trains another NER model on the pseudo-annotated corpus. The difference is that PredictMine does not use our divergence distance, instead, it uses model predictions from an NER model trained on 1-shot data;

- **CertMine** is similar to PredictMine but additionally includes a sequential uncertainty estimation of predicted entities by calculating the token-wise average of negative log probability on each predicted span. CertMine only pseudo-annotates entities with the lowest $10\%$ uncertainty.

Our X-NER by default uses `roberta-large`. For all the compared methods, we use a comparable size, if not more favorable, language model for them (e.g., `bert-large`, `bart-large`, `GPT-XL`).

### 5.3 NER Performance Comparison

Table 2 presents the end-to-end NER evaluation results of different methods that operate under the XWS setting. While arguably PromptMine requires one less entity example than X-NER, its performance is much subpar compared to ours on all four datasets. When compared with

### 5.4 Span Mining Performance Analysis

To better understand why our divergence metric is better in mining entities, we conduct a fine-grained

---

[4]The experiments are conducted in the timeframe May 23rd - June 6th, which corresponds to the 0313 version

analysis.

**Span Ranking Method Comparison** Table 4 presents the performance of various XWS methods for entity mining on the NER datasets, measured using Precision@1000. The results reveal that X-NER significantly outperforms other methods across all datasets and shows remarkably high precision on specific entity types, particularly PER and MISC entities.

In the 2-way configuration with divergence distance, X-NER outperforms other setups such as 1-way span testing and the cosine similarity metric. This verifies the mining precision to provide solid support to X-NER's performance. X-NER also achieves an outstanding Precision@1000 score of 98.4 for PER and 97.9 for LOC entities in the CoNLL03 dataset. Comparatively, other methods such as CertMine, PromptMine, and CosMine exhibit lower performance. CertMine achieves an average Precision@1000 of 16.5, while PromptMine performs better with average scores of 22.9 (GPT2-XL) and 24.3 (RoBERTa-L). However, these scores still fall significantly short of X-NER's performance.

**Span Extraction Method Comparison** As illustrated in Figure 3, we use precision-recall graphs to compare our model's performance in a 1-shot setting with NER extractors. When compared to PredictMine, X-NER performs similarly to a model trained with $10 \sim 20$ shots, except for the ORG category. This can be attributed to the challenging diversity of ORG entities. In comparison to ChatGPT, X-NER demonstrates competitive performance on LOC entities and superior performance on MISC entities. A single-shot example isn't enough to guide ChatGPT to extract the correct entity phrases, due to the difficulty in switching its memorized concept of MISC. However, ChatGPT maintains high performance in PER and ORG categories, thanks to its comprehensive knowledge of entities. For these labels, X-NER's advantage lies in its ability to maintain high precision while using a smaller sampling rate.

## 6   Case Studies

### 6.1   Span Coverage and Subset Decomposition

As X-NER is a similarity-based method, we illustrate how we can use X-NER to *discover* fine-grained entity types of a given coarse-grained entity

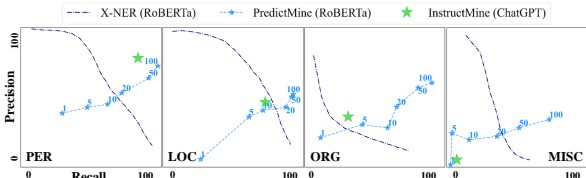

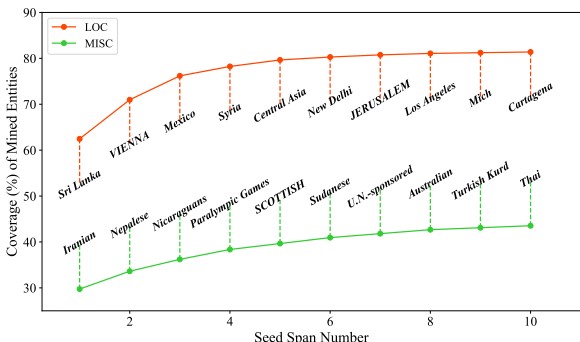

Figure 3: Comparison with k-shot tuning and ChatGPT.

Figure 4: Fine-grained entity results on LOC and MISC in CoNLL03. The entities listed in order from left to right are decomposed fine-grained entity types for location and miscellaneous.

type.[5]

We present an example of decomposing LOC from CoNLL03, in which we find a few representative fine-grained location entities. We first sample 1000 sentences with their ground truth annotations from the CoNLL03 dataset and denote them by $\mathcal{E}$. Next, we randomly sample 50 seed LOC entities from $\mathcal{E}$; the goal is to identify a few representative entities from these 50 entities. For each sampled entity $e_i$, we use it as a reference to mine entities from $\mathcal{E}$ using X-NER. Because of fine-grained awareness of X-NER, the mined entities with the lowest distance are not only location entities, but also locations of a similar type as $e_i$. We denote $E_i$ as the set of mined entities.

Then, we employ a maximal coverage algorithm to help us identify the unique $E_i$'s that span over the location entities in $\mathcal{E}$. We simply use a greedy algorithm that progressively selects $E_i$ that contains the most entities that were not contained in previous selections.

The results are shown in Figure 4, where we additionally include an experiment with the MISC entity type. By looking at the first selected entities $e_i$ for Location, we can find different fine-grained entity types. For instance, the first five seed spans for

---

[5]More discussion about fine-grained awareness can be found in Appendix D.

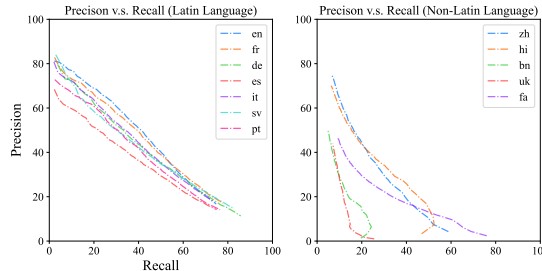

Figure 5: Cross-lingual span mining Performance of X-NER on the MultiCoNER training dataset.

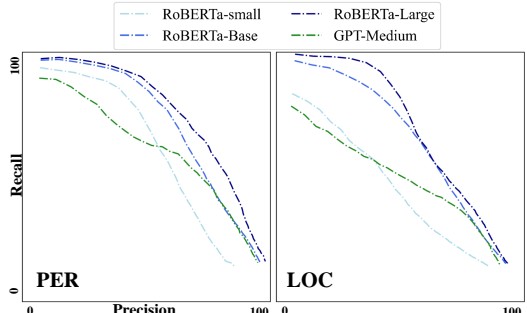

Figure 6: The correlation between model scale and mining ability.

the LOC category include the country (*Sri Lanka*, *Mexico*, *Syria*), city (*VIENNA*), and region (*Central Asia*). Thus, we conclude that our similarity evaluation successfully decomposes coarse-grained categories into fine-grained ones.

## 6.2 Cross-Lingual Results

Our entity mining technique holds promising potential as a useful tool for languages with fewer resources. This is because it can efficiently extract entities within a single span, eliminating the need for expert annotators. Furthermore, we've illustrated how our method can be successfully applied in cross-lingual contexts. For example, as shown in Figure 5, we utilized an English span to extract entities in various other languages, leveraging the cross-lingual masked language model, XML-Roberta-L. We test the quality of pseudo entities labeled in the cross-lingual dataset.

During our comprehensive experimental process, we choose to utilize the MultiCoNER (Malmasi et al., 2022) dataset, which encompasses data from 12 languages, as a platform to extract multilingual person entities. Our choice of the dataset was driven by its diversity and comprehensiveness, enabling us to test our approach across a broad range of linguistic contexts. When compared to English, X-NER demonstrates remarkable resilience and effectiveness in most Latin-based languages. This proved the robustness of our entity mining technique, reinforcing its potential as a universally applicable tool in a broad array of linguistic environments. While the performance is somewhat reduced in non-Latin languages, our X-NER still manages to identify relevant entities, showcasing the universality of our technique.

## 6.3 Backbone Analysis

We compare different backbone models and present their performances in Figure 6.

**Ability vs. Model Size** The figure highlights that the entity mining ability of our X-NER commensurately improves with the backbone model size. It's noteworthy that even models of *small* size display a decent mining capacity, suggesting that more compact models can serve as viable starting points for this task. As these models grow into *base* and *large* sizes, there is a noticeable enhancement in their mining ability. This indicates a direct, positive correlation between model size and extraction efficiency, validating the scalability of our method and potentially guiding future development with larger models. Furthermore, our analysis reveals a consistency in performance rankings across different entity types, irrespective of model scale. This suggests that some entities may inherently present more challenges for extraction than others, a consistent factor that could inform future strategies for mining optimization.

**MLM vs. CLM** In our experimental setup, we generally employ bidirectional PLMs as opposed to their unidirectional counterparts. This strategic decision was primarily motivated by our belief that bidirectional context information is of paramount importance when assessing divergence distance, a critical factor in our methodology. This section aims to draw a comparison between the information extraction capabilities of these two types of PLMs. For the unidirectional, CLM-based GPT2, we adapted our label context to follow the structure of "[SEED SPAN] is a/an [ENTITY LABEL] name" to make it compatible with the unidirectional context understanding. The results showed the MLM-based `roberta-large` (355M) to considerably outperformed the CLM-based `gpt2-medium` (345M). This marked performance difference lends support to our initial hypothesis, demonstrating the inherent advantage of bidirectional language modeling in

| Method | Sampling | Training | Inference |
|---|---|---|---|
| Tagger | - | $O(N)$ | $O(N)$ |
| PredictMine | $O(N)$ | $O(N)$ | $O(N)$ |
| CertMine | $O(N)$ | $O(N)$ | $O(N)$ |
| TemplateNER | - | $O(N)$ | $O(NL)$ |
| EntLM | - | $O(N)$ | $O(N)$ |
| SpanNER | - | $O(N)$ | $O(N)$ |
| FFF-NER | - | $O(NL)$ | $O(NL)$ |
| X-NER | $O(NL)$ | $O(N)$ | $O(N)$ |

Table 5: Time complexity analysis of different few-shot NER frameworks.

| Method | #Instance | #Label | Time |
|---|---|---|---|
| CoNLL03 | $14.0K$ | 4 | $3.7H$ |
| Tweebank | $1.6K$ | 4 | $0.5H$ |
| WNUT17 | $3.4K$ | 6 | $1.7H$ |
| Restaurant | $7.7K$ | 8 | $1.8H$ |

Table 6: The time cost of X-NER for sampling on different training datasets.

effectively capturing context and improving entity extraction.

### 6.4 Time Complexity

We present a time complexity analysis that compares the time consumed by different methods in Table 6, which will be added to the new version of our paper. $N$ refers to the number of instances for sampling/training/inference. $L$ refers to the length of sentences, where the average length can be used for approximation.

From the table above we can see our X-NER is as efficient as a normal tagger and faster than some baselines like FFF-NER during inference since we train a tagger based on the mined corpus. The main time burden for X-NER is from mining where we test about $3L$ spans for $N$ instances in the unlabeled corpus.

### 7 Conclusions and Future Work

In summary, our research offers a groundbreaking Extremely Weak Supervision method for Named Entity Recognition, which significantly enhances 1-shot NER performance. We generate high-quality pseudo annotations without requiring fine-tuning by capitalizing on pre-trained language models' inherent text comprehension. Additionally, we address challenges related to span representation and context integration with a novel span test, leading to superior results on NER tasks compared to many established methods. Future work will focus on improving the scalability of this approach, enhancing

its efficacy across a broader array of NER scenarios, and refining the span test for more nuanced linguistic contexts.

### Limitations

While our proposed method, X-NER, has shown commendable performance in NER under an extremely weak supervision setting, it does come with certain limitations. Firstly, the time complexity of generating pseudo-labels is significant, largely due to the need for similarity computation between the context distributions of entity spans and example entities in an unlabelled training corpus. This makes the process less time-efficient, particularly in cases of extensive training data. Secondly, our approach heavily depends on the understanding of the PLM used. If the PLM fails to adequately comprehend or interpret the entity, the method might fail to generate effective pseudo-labels, leading to less accurate NER tagging. Lastly, our method may sometimes wrongly classify entity spans that are encapsulated within another entity. For instance, in the case of the entity "New York University", the method might incorrectly identify "New York" as a separate location entity. Despite these challenges, our method demonstrates substantial promise, though it points to future work for refining the approach to mitigate these limitations.

### Ethical Consideration

Our work focuses on the traditional NER task, which generally does not raise ethical consideration.

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

## A Seed Entity and Template

| | Entity Label | Seed Span |
|---|---|---|
| **CoNLL03** | PER | Masaki Yoshida |
| | LOC | Singapore |
| | ORG | Boston Celtics |
| | MISC | Japanese |
| **Tweebank** | PER | Masaki Yoshida |
| | LOC | Singapore |
| | ORG | Boston Celtics |
| | MISC | Japanese |
| **WNUT17** | Person | Masaki Yoshida |
| | Location | Singapore |
| | Creative work | #BattlestarGalactica |
| | Group | Boston Celtics |
| | Corporation | Microsoft |
| | Product | iPhone |
| **Restaurant** | Restaurant name | McDonald's |
| | Location | nearby |
| | Dish | chicken sandwich |
| | Price | cheap |
| | Rating | five star |
| | Cuisine | Japanese |

Table 7: Entity Labels and Seed Spans

| Template |
|---|
| Based on these examples: |
| <example> |
| Bracket and label the named entities in the sentence: |
| <sentence> |

Table 8: The XWS template used for ChatGPT in InstructMine.

## B Dataset Statistics

The statistics of the datasets in our experiments are presented in Table 9.

## C Pseudo Dataset Construction

We also include several constraints to construct a high-quality pseudo dataset. These constraints are applied to the two span testing-based methods: PromptMine and X-NER. First, we only test sentences under certain lengths (token numbers). Since each discovered entity will add the whole

| Dataset | CoNLL03 | Tweebank | WNUT17 | Restaurant |
|---|---|---|---|---|
| Domain | News | Social Media | Social Media | Review |
| #Train | $14.0K$ | $1.6K$ | $3.4K$ | $7.7K$ |
| #Test | $3.5K$ | $1.2K$ | $1.3K$ | $1.5K$ |
| #Label | 4 | 4 | 6 | 8 |

Table 9: The statistics of the NER dataset in our experiments.

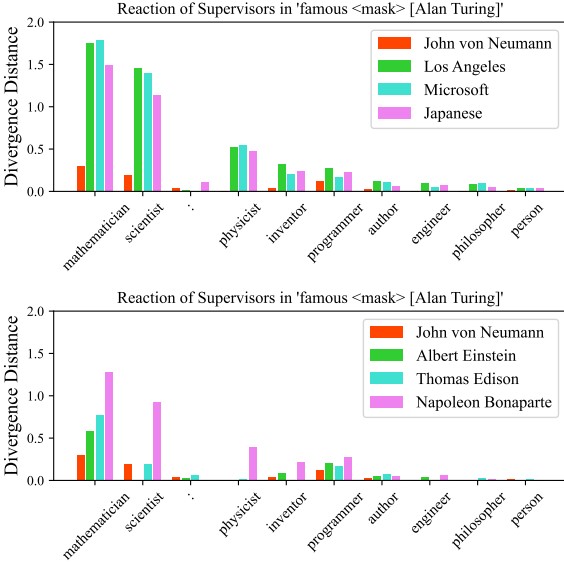

Figure 7: Supervisor reaction to span test for coarse and fine-grained entities.

sentence into the pseudo dataset, we avoid too-long sentences which more likely suffer from missing annotation. The maximal length threshold for CoNLL03 and Tweebank is 30, while for WNUT17 and Restaurant is 20.

For the similarity threshold, we keep it constant for different labels in the same data. For divergence distance, the threshold for CoNLL03 and WNUT17 is $< 0.15$, while for Tweebank and Restaurant is $< 0.20$. This policy is also applicable to cosine similarity where the threshold is $> 1.999$. For PromptMine, since the score is not evenly distributed, we label the top $2\%$ span for each entity category.

## D Fine-grained Interpretation

**Reaction of Neighbors (Supervisors)** We further explore how the pre-trained LM understands the semantics of the seed entity. In Figure 7, we present case studies demonstrating how the masked predictions on neighbor words (supervisors) change when *Alan Turing* is substituted with a few selected entities of the same coarse-grained. Here, supervisor-wise distance refers to the weighted divergence $p(x_k =$

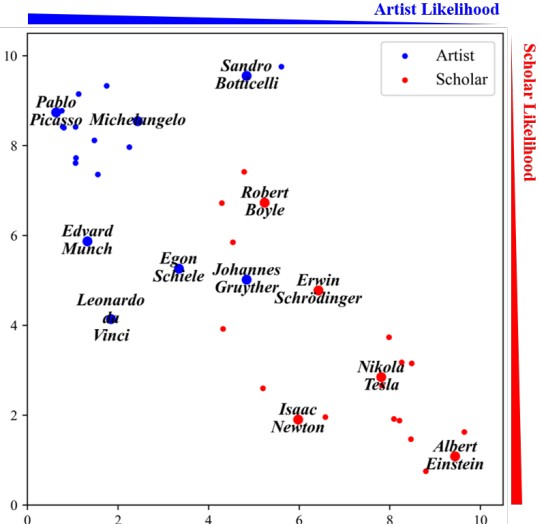

Figure 8: Decomposed probability distributions. **Negative log probability** on axes.

$v|X)\frac{p(x_k=v|X)}{p(x_k=v|X_{[i:j\to Z]})}$ of supervisor $v$. We plot the distance of supervisors $v$ with the top-$K$ weight $p(x_k = v|X)$.

For entities with distinct coarse-grained categories, most supervisors are activated to identify the differences, showcasing their strong ability for coarse-grained categorization. For other PER entities, a supervisor is activated only when the entities differ in the attribute it represents. For example, *Albert Einstein* activates the *mathematician* supervisor but not the *scientist* supervisor. Conversely, *Thomas Edison* activates both the *mathematician* and *scientist* supervisors but not the *inventor* supervisor. Interestingly, *Napoleon Bonaparte*, the PER entity most dissimilar to *Alan Turing*, activates the most supervisors, highlighting the sensitivity of supervisors to fine-grained differences among entities.

**Decomposed Divergence Distance** We further explore how the performance of our X-NER is supported by the pre-trained LM's understanding of named entities. By examining the varying responses of supervisors to different entities, we can break down the divergence distance for each supervisor. This decomposition is demonstrated for *artist* and *scholar* in Figure 8. In this case, we select 20 renowned artists and scholars, positioning them based on the negative log probability of the *artist* and *scientist* predictions from the *famous <mask> [NAME]* prompt. Interestingly, we can observe that the artists and scholars automatically cluster into two distinct groups, showcasing X-

NER's ability to distinguish fine-grained entities. Notably, *Leonardo da Vinci*, who is both an artist and a scholar, is positioned closest to the origin, indicating that X-NER is effectively capturing the dual nature of Leonardo da Vinci's expertise.