# OpenReview forum: "Less than One-shot: Named Entity Recognition via Extremely Weak Supervision"
_EMNLP/2023/Conference — EMNLP 2023 Findings_

### Official Review · Reviewer_anZR · 2023-08-01

**Soundness:** 3

**Excitement:**

3: Ambivalent: It has merits (e.g., it reports state-of-the-art results, the idea is nice), but there are key weaknesses (e.g., it describes incremental work), and it can significantly benefit from another round of revision. However, I won't object to accepting it if my co-reviewers champion it.

**Missing References:**

This is a small subset of existing works.
* Few-Shot Named Entity Recognition: An Empirical Baseline Study
* CONTAINER: Few-Shot Named Entity Recognition via Contrastive Learning
* Simple and Effective Few-Shot Named Entity Recognition with Structured Nearest Neighbor Learning
* Few-shot Named Entity Recognition with Self-describing Networks
* SEE-Few: Seed, Expand and Entail for Few-shot Named Entity Recognition
* A Little Annotation does a Lot of Good: A Study in Bootstrapping Low-resource Named Entity Recognizers

**Paper Topic And Main Contributions:**

This paper studies NER under the extremely weak supervision setting, where only one example per entity type is given in the "The [Class LABEL]; <example>" structure.
The proposed system (X-NER)  uses the one-shot examples to obtain high-quality pseudo labels from an unlabeled data using a pre-trained LM and train an NER model using the standard sequence tagging method.

The steps to generate pseudo-labels using a large unlabelled data and one example per entity type:
    (1) It finds a candidate span in a sentence for the target entity type
    (2) Swap the span with the example entity
    (3) Measure the LM’s prediction distribution of the tokens in the context and computer the KL divergence between the contexts before and after the swap
    (4) Rank all spans and select top-K entities to construct pseudo labels for the entity type

The main contribution is to use the KL divergence of the context words, instead of the cosine similarity between entity spans, to choose new entity mentions from unlabeled sentences to generate pseudo labeled sample. Their experiments and ablation studies show the effectiveness of this approach.

**Questions For The Authors:**

** In line 315, “We don’t view this as additional supervision, but rather akin to an instruction template used in prompting methods “ ==> This statement is debatable. Prompts are a form of additional supervision. Not labeled does not mean they are not additional information.

** In Table 3, "Due to a limitation of resources, we run the experiments only on  the standard CoNLL03 benchmark."    Can you be more specific about the resource limitation preventing doing 1-shot experiment for other datasets? It seems to be easy to do. For instance, you can take one sentence for each entity type from the train or dev split.

** In Table 3, : Do PredictMine and CertMine (slight variations of X-NER) use the same unlabeled corpus and K as X-NER to generate pseudo labels?

** It is more conventional to call the provided example "seed example" rather than  "seed span" as it is just a term not a sentence.

**Reasons To Accept:**

** Few-shot NER is an important area to apply NER in real-world problems.
** The use the KL divergence between the context words before and after replacing a candidate span with the seed example to decide the likelihood of the span belonging to the same category as the seed example is new and promising.
** The paper shows several ablation experiments providing good insights.

**Reasons To Reject:**

** The idea is few (one)-shot NER has been studied by many researchers both with and without LLMs (See missing references below) . The comparisons provided in this paper are not sufficient.  This paper compares X-NER against pure few-shot methods (e.g, FFF-NER) and non few-hot method (SpanNER) but does not compare X-NER method against existing methods which also utilize unlabeled data.

** The paper does not provide important details on how X-NER works. (1) how do you generate  the candidate spans to swap with the seed entity? do you consider all possible spans up to a certain length as in SpanNER or do you use some heuristics?  (2) how did you construct the unlabeled corpus for the experiments? the performance of X-NER will depend on the quality of the unlabeled data. (3) what was the value of K in top-K entities selected by the entity miner?

** The argument of using only one example is not convincing, as providing a few examples with the class name is very easy to do in practice.

**Reproducibility:**

3: Could reproduce the results with some difficulty. The settings of parameters are underspecified or subjectively determined; the training/evaluation data are not widely available.

**Reviewer Confidence:**

4: Quite sure. I tried to check the important points carefully. It's unlikely, though conceivable, that I missed something that should affect my ratings.

---

> ### Author Rebuttal · Authors · 2023-08-28
>
> Thank you for showing enthusiasm towards our research! We value your inquisitiveness and look forward to diving further into our framework for named entity recognition with minimal supervision (XWS). To bolster our claims, we have added the comparison with more baselines and provided a detailed breakdown of our implementation.
>
> - **More Comparisons and References**
>
> We have added the comparison with your mentioned works. We are also going to add descriptions and citations of those works in the related works section.
>
> | Method     | P.     | R.     | F.     |
> |------------|-------|-------|-------|
> | **XWS**        |        |        |       |
> | X-NER      | 71.4  | 52.1  | 60.2  |
> | **1-shot**     |        |        |       |
> | Tagger    | 17.2  | 12.8  | 14.6  |
> | PredictMine  | 13.5  | 17.2  | 15.1  |
> | CertMine    | 20.2  | 14.7  | 17.0  |
> | **ETAL**      | 26.4  | 17.1  | 20.8  |
> | **SEE-Few**   | 21.5  | 26.7  | 23.8  |
> | TemplateNER | 12.6  | 6.7   | 8.8   |
> | EntLM     | 13.6  | 29.0  | 18.5  |
> | SpanNER   | 56.1  | 3.8   | 7.1   |
> | FFF-NER   | 39.1  | 3.6   | 6.5   |
> | **SDNET**   | 56.6  | 44.1  | 51.3  |
> | **5-shot**    |        |        |       |
> | Tagger    | 58.9  | 55.1  | 56.9  |
> | FFF-NER   | 70.2  | 65.3  | 67.7  |
>
> **Bold Methods** are newly added to the next version. **ETAL** is a method with pseudo-labeling to search for highly-confident entities that maximize the probability of BIO sequences. **SEE-Few** expands seeded entities and applies an entailment framework to efficiently learn from a few examples. **SDNET** pre-trains a T5 on silver entities from Wikipedia and then fine-tunes it on few-shot examples.
>
> Our results add more distantly-supervised and self-training baselines and our X-NER significantly outperforms them. Remarkably, our method outperforms SDNET, which uses pre-trained models on silver NER labels from Wikipedia.
>
> - **Implementation Details**
>
>  (1) For swapping candidates, we mention in \textbf{5.1} that all spans with no more than $3$ tokens and without stop words are selected for each sentence. (2) The construction of the pseudo-labeling dataset is in \textbf{Appendix C}, we set a limitation of similarity and length for datasets. (3) The $K$ is determined by a similarity threshold that is also mentioned in \textbf{Appendix C} and we will move these into the main paper in the updated version. (4) You are correct that prompts usually provide extra supervision for the framework, but the prompt in our work is simply "<label name>: <span example>", which provides very little extra supervision. (5) In Table 3, as 1-shot methods clearly lag behind our X-NER, running all compared methods on all the datasets takes a lot of GPU resources, with little additional value. (6) **PredictMine** and **CertMine** use exactly the same unlabeled corpus (the corresponding training dataset for each test dataset) and the same $K$ as our X-NER. (7) Your suggestion on replacing "seed example" with "seed span" clearly captures the meaning of this element in our experiment, we apply the replacement to the version of our paper.
>
> - **Multiple Seed Examples**
>
> We study single seed examples rather than multiple ones due to two reasons: (1) Single seed example has lesser burden in providing supervision lessens users effort. One can definitely explore the case where more supervision is allowed, building upon our results. (2) Single seed example is the arguably minimum supervision necessary for a human to solve NER, so we design a method to make machines work under this setting. (3) As an example of incorporating multiple seed examples, we experiment with pseudo-labeling with multiple seed examples by averaging the similarity scores. Experiment results show our X-NER benefits from more seed examples. For instance, using two person seed examples "Masaki Yoshida'' and "John Smith" will achieve a better precision-recall curve on CoNLL03 than only using ``Masaki Yoshida'' in our current experiments. We will add this result to the next version of our paper.

---

### Official Review · Reviewer_o4z4 · 2023-08-07

**Soundness:** 4

**Excitement:**

4: Strong: This paper deepens the understanding of some phenomenon or lowers the barriers to an existing research direction.

**Paper Topic And Main Contributions:**

This paper targets the extremely weak supervision (XWS) setting of named entity recognition (NER), a setting that is rarely studied in previous research. One core step of the proposed approach is the construction of pseudo-labels from the training corpus. The authors propose a smart way to perform span test based on language models, by comparing the context distributions before and after the span is replaced by the entity example.

**Reasons To Accept:**

Strengths:
1. The proposed approach is novel and effective.
2. It is amazing that the proposed method for the XWS setting outperforms many methods designed for 1-shot or few-shot settings.
3. The experiments are solid, and the analysis is convincing and informative.

**Reasons To Reject:**

The paper appears to be free of significant weaknesses.

**Reproducibility:**

4: Could mostly reproduce the results, but there may be some variation because of sample variance or minor variations in their interpretation of the protocol or method.

**Reviewer Confidence:**

4: Quite sure. I tried to check the important points carefully. It's unlikely, though conceivable, that I missed something that should affect my ratings.

---

> ### Author Rebuttal · Authors · 2023-08-28
>
> We are sincerely thankful for your appreciation of our work! You can refer to our responses to other reviewers to view the updates we make to further improve the quality of the work.

---

### Official Review · Reviewer_c5pW · 2023-08-10

**Soundness:** 3

**Excitement:**

3: Ambivalent: It has merits (e.g., it reports state-of-the-art results, the idea is nice), but there are key weaknesses (e.g., it describes incremental work), and it can significantly benefit from another round of revision. However, I won't object to accepting it if my co-reviewers champion it.

**Paper Topic And Main Contributions:**

This paper presents a challenging Extremely Weak Supervision NER setting, i.e., only one entity mention is provided for a entity type, and proposes a pseudo label method to resolve it. This paperutilizes the ability of language model to contextually model and obtain pseudo labels from unlabeled text, forming training data. Experimental results show that the method proposed in this paper has demonstrated excellent performance, surpassing both the 1-shot approach and the method of generating pseudo-labels through chatgpt.

**Questions For The Authors:**

QA Will different seed spans have significant differences in results?

QB Why does two-way have such a significant impact on MISC and ORG in Table 4?

**Reasons To Accept:**

A. This paper presents a challenging Extremely Weak Supervision NER setting.

B. The method described in this paper is reasonable and has achieved excellent results.

C. Experimental results have demonstrated the excellent cross-lingual effectiveness of the method proposed in this paper.

**Reasons To Reject:**

A. The experiment in Table 3 seems unfair because compared to incorporating pseudo-labeled data, 1-shot fine-tuning clearly performs worse.

B. The setting and methodology of this article are similar to distant supervision and self-training, but the baseline lacks comparison with such methods.

C. The method described in this paper seems to be time-consuming and lacks a comparison of the time spent on each method.

D. This paper did not analyze the potential impact of different seed spans.

E. There is no mention of how to set a similarity threshold for a new dataset.

**Reproducibility:**

4: Could mostly reproduce the results, but there may be some variation because of sample variance or minor variations in their interpretation of the protocol or method.

**Reviewer Confidence:**

3: Pretty sure, but there's a chance I missed something. Although I have a good feel for this area in general, I did not carefully check the paper's details, e.g., the math, experimental design, or novelty.

---

> ### Author Rebuttal · Authors · 2023-08-28
>
> Thank you for expressing interest in our work! We appreciate your curiosity and are excited to delve deeper into our framework for named entity recognition under extremely weak supervision. We have incorporated more experiments and comparisons to support the claims and contributions of our work.
>
> - **Pseudo-labeling Baselines (For A, B)**
>
>  For pseudo-labeling baselines, we have already included mining with a predictor **PredictMine** and **CertMine** and variants with cosine similarity or prompting probability (**PromptMine**) in the current version. We are also glad to add more distantly-supervised or self-training baselines in the following table.
>
> | Method     | P.     | R.     | F.     |
> |------------|-------|-------|-------|
> | **XWS**        |        |        |       |
> | X-NER      | 71.4  | 52.1  | 60.2  |
> | **1-shot**     |        |        |       |
> | Tagger    | 17.2  | 12.8  | 14.6  |
> | PredictMine  | 13.5  | 17.2  | 15.1  |
> | CertMine    | 20.2  | 14.7  | 17.0  |
> | **ETAL**      | 26.4  | 17.1  | 20.8  |
> | SEE-Few   | 21.5  | 26.7  | 23.8  |
> | TemplateNER | 12.6  | 6.7   | 8.8   |
> | EntLM     | 13.6  | 29.0  | 18.5  |
> | SpanNER   | 56.1  | 3.8   | 7.1   |
> | FFF-NER   | 39.1  | 3.6   | 6.5   |
> | **SDNET**   | 56.6  | 44.1  | 51.3  |
> | **5-shot**    |        |        |       |
> | Tagger    | 58.9  | 55.1  | 56.9  |
> | FFF-NER   | 70.2  | 65.3  | 67.7  |
>
> **Bold Methods** are newly added to the next version. **ETAL** is a method with pseudo-labeling to search for highly-confident entities that maximize the probability of BIO sequences. **SEE-Few** expands seeded entities and applies an entailment framework to efficiently learn from a few examples. **SDNET** pre-trains a T5 on silver entities from Wikipedia and then fine-tunes it on few-shot examples.
>
> Our results add more distantly-supervised and self-training baselines and our X-NER significantly outperforms them. Remarkably, our method outperforms SDNET, which uses pre-trained models on silver NER labels from Wikipedia.
>
> - **Time Complexity (For C)**
>
> We add a time complexity analysis that compares the time consumed by different methods in the following table, which will be added to the new version of our paper. $N$ refers to the number of instances for sampling/training/inference. $L$ refers to the length of sentences, where the average length can be used for approximation.
>
> | Method      | Sampling | Training | Inference |
> |-------------|----------|----------|-----------|
> | Tagger      | -        | O(N)     | O(N)      |
> | PredictMine | O(N)     | O(N)     | O(N)      |
> | CertMine    | O(N)     | O(N)     | O(N)      |
> | TemplateNER | -        | O(N)     | O(NL)     |
> | EntLM       | -        | O(N)     | O(N)      |
> | SpanNER     | -        | O(N)     | O(N)      |
> | FFF-NER     | -        | O(NL)    | O(NL)     |
> | X-NER       | O(NL)    | O(N)     | O(N)      |
>
> From the table above we can see our X-NER is as efficient as a normal tagger and faster than some baselines like FFF-NER during inference since we train a tagger based on the mined corpus. The main time burden for X-NER is from mining where we test about $3L$ spans for $N$ instances in the unlabeled corpus.
>
> | Method      | #Instance | #Label | Total Time  |
> |-------------|-----------|--------|-------|
> | CoNLL03     | 14.0K     | 4      | 3.7H  |
> | Tweebank    | 1.6K      | 4      | 0.5H  |
> | WNUT17      | 3.4K      | 6      | 1.7H  |
> | Restaurant  | 7.7K      | 8      | 1.8H  |
>
> The result is from runs on a single RTX-3090. The time costs are controlled under 4 hours on all datasets, which verifies our X-NER to be an applicable method for NER.
>
> - **Seed Span Analysis (For D and QA)**
>
> We present an analysis of the impact of different seed examples. As we cannot present a precision-recall curve on OpenReview, we describe the setting and results from our experiments. We test four seed examples "Masaki Yoshida", "John Smith", "Haruki Murakami", and "Alan Turing". We find the seed examples do not have a significant impact on the precision-recall curve. Additionally, general names like "Masaki Yoshida" and "John Smith" outperform famous names, which might be attributed to their less bias to entities with similar properties like career and title.
>
> - **Similarity Threshold (For E)**
>
> Based on our experiments, we find a threshold between $0.15$ to $0.20$ works well across all tested datasets for mining on the unlabeled corpus. Thus, one can try these two thresholds when facing a new dataset.
>
> - **Two-way Testing (For QB)**
>
> The most significant improvement of two-way testing on ORG and MISC can be attributed to the contextual diversity of these entities and the replacement of tested spans to seed examples provides a fair context to calculate the distance between them.

---

### Meta-Review · Area_Chair_aTch · 2023-09-22

**Recommendation:** 4

**Metareview:**

Reviewers recognized the contribution of this paper to low resource NER settings, and the efficacy of the proposed data augmentation on NER benchmarks. The reviewers also suggested additional baseline comparisons and related work which provide important comparisons for the specific application, which were added during the rebuttal discussion. Reviewers also had several questions about the methodology, scalability, and more precise description of the approach which should be addressed in revision. This paper will likely be of moderate interest to the EMNLP community with contributions to few-shot learning and data augmentation strategies.

---

### Decision · Program_Chairs · 2023-10-07

**Decision:**

Accept-Findings

**Comment:**

Reviewers recognized the contribution of this paper to low resource NER settings, and the efficacy of the proposed data augmentation on NER benchmarks. The reviewers also suggested additional baseline comparisons and related work which provide important comparisons for the specific application, which were added during the rebuttal discussion. Reviewers also had several questions about the methodology, scalability, and more precise description of the approach which should be addressed in revision. This paper will likely be of moderate interest to the EMNLP community with contributions to few-shot learning and data augmentation strategies.